# PONDERLM: PRETRAINING LANGUAGE MODELS TO PONDER IN CONTINUOUS SPACE

**Boyi Zeng**[1], **Shixiang Song**[1,2,3], **Siyuan Huang**[1], **Yixuan Wang**[1,3], **He Li**[1],
**Ziwei He**[3],**Xinbing Wang**[4], **Zhiyu Li**[2], **Zhouhan Lin**[1,2,3]*
[1]LUMIA Lab, Shanghai Jiao Tong University,
[2]Institute for Advanced Algorithms Research, Shanghai,
[3]Shanghai Innovation Institute, [4]Shanghai Jiao Tong University
boyizeng@sjtu.edu.cn *lin.zhouhan@gmail.com

## ABSTRACT

Humans ponder before articulating complex sentence elements, enabling deeper cognitive processing through focused effort. In this work, we introduce this pondering process into language models by repeatedly invoking the forward process within a single token generation step. During pondering, instead of generating an actual token sampled from the prediction distribution, the model ponders by yielding a weighted sum of all token embeddings according to the predicted token distribution. The generated embedding is then fed back as input for another forward pass. We show that the model can learn to ponder in this way through self-supervised learning, without any human annotations. Experiments across three widely used open-source architectures—GPT-2, Pythia, and LLaMA—and extensive downstream task evaluations demonstrate the effectiveness and generality of our method. On 9 downstream benchmarks, our pondering-enhanced Pythia models significantly outperform the official Pythia models. Notably, our PonderPythia models demonstrate remarkable effectiveness: PonderPythia-2.8B surpasses Pythia-6.9B and rivals Pythia-12B, while our PonderPythia-1B matches TinyLlama-1.1B, a model trained on 10 times more data.[1]

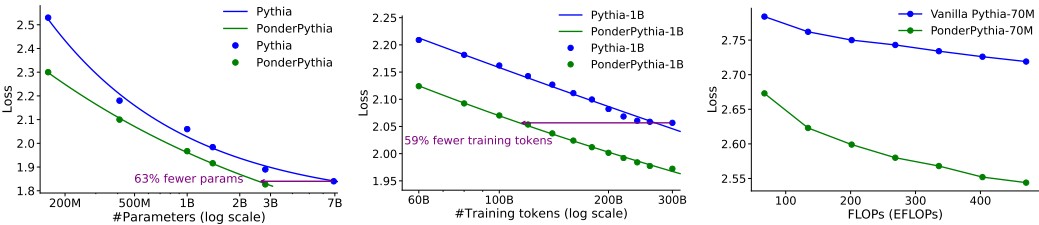

Figure 1: Scaling curves comparing PonderPythia with the official Pythia suite on the 300B Pile. Our 2.55B model matches the loss of Pythia-6.9B with 63% fewer parameters (left), while our PonderPythia-1B model reaches the baseline's final performance with 59% less training data (middle). Furthermore, for the same computational budget (FLOPs), PonderPythia consistently achieves a lower loss than the baseline (right).

## 1 INTRODUCTION

In the pursuit of improving model performance, scaling up model parameters and data sizes has long been the most widely adopted and accepted approach (Kaplan et al., 2020; Brown et al., 2020; Liu et al., 2024). However, this approach faces several bottlenecks, including the exhaustion of high-quality data (Villalobos et al., 2022; Muennighoff et al., 2023), the observed saturation in scaling laws (Hackenburg et al., 2025; Hoffmann et al., 2022a) and substantial communication overhead in

---

*Zhouhan Lin is the corresponding author.
[1]The code is available at https://github.com/LUMIA-Group/PonderingLM.

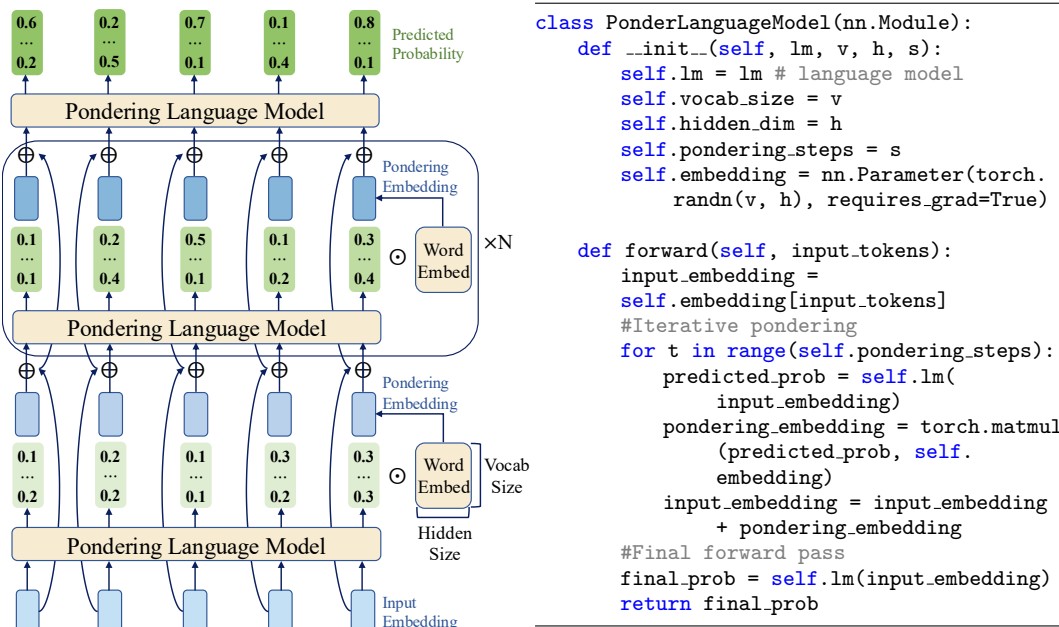

```python
class PonderLanguageModel(nn.Module):
    def __init__(self, lm, v, h, s):
        self.lm = lm # language model
        self.vocab_size = v
        self.hidden_dim = h
        self.pondering_steps = s
        self.embedding = nn.Parameter(torch.
            randn(v, h), requires_grad=True)

    def forward(self, input_tokens):
        input_embedding =
        self.embedding[input_tokens]
        #Iterative pondering
        for t in range(self.pondering_steps):
            predicted_prob = self.lm(
                input_embedding)
            pondering_embedding = torch.matmul
                (predicted_prob, self.
                embedding)
            input_embedding = input_embedding
                + pondering_embedding
        #Final forward pass
        final_prob = self.lm(input_embedding)
        return final_prob
```

Figure 2: Overview of the PonderLM. Given input token embeddings, the base LM produces a probability distribution over the vocabulary, which is used to compute a continuous "pondering embedding" via a weighted sum of all token embeddings. This embedding is then added residually to the original input embeddings and fed back into the LM. By repeating this process for $k$ steps within a single token prediction, the model iteratively refines its output distributions. The pseudocode on the right illustrates the implementation details.

distributed pre-training that grows super-linearly with model size (Narayanan et al., 2021; Pati et al., 2023; Li et al., 2024).

On the other hand, if we look at humans, the growth of human capabilities does not stem from simply increasing the number of neurons in the brain. Instead, when faced with complex problems, humans often enhance their problem-solving abilities by repeatedly pondering, engaging in deep cognitive processing before articulating their thoughts.

Analogously, in large language models, the most relevant research direction is test-time scaling. In particular, following the advancements in o1 and R1 (Jaech et al., 2024; DeepSeek-AI et al., 2025), generating long chains of thought (CoT) has emerged as the mainstream approach for scaling test-time computation. However, CoT-based methods also exhibit several drawbacks: they often require curated human-annotated datasets (Allen-Zhu & Li, 2023) and carefully designed reinforcement learning algorithms (Pang et al., 2025). Moreover, small models rarely benefit from CoT (Li et al., 2023), and the upper bound of performance remains constrained by the base pretrained model (Yue et al., 2025). Additionally, current language models employing CoT are still confined to discrete language spaces with fixed vocabularies, which, according to recent studies (Fedorenko et al., 2024; Hao et al., 2024; Pfau et al., 2024), are primarily optimized for communication rather than for internal computational thinking.

To overcome these challenges and inspired by human pondering, we introduce the Pondering Language Model (PonderLM), which relies solely on self-supervised learning. PonderLMs can be naturally learned through pretraining on large-scale general corpora, without the need for human-annotated datasets or reinforcement learning.

During pondering, instead of generating a discrete token sampled from the prediction distribution, the model produces a weighted sum of all token embeddings based on the predicted probabilities. This generated embedding is then fed back into the language model, allowing it to iteratively refine its predictions. As the weighted embedding is continuous, PonderLMs overcome the expressive limitations of discrete token vocabularies and enable fully differentiable, end-to-end pretraining via gradient descent. Furthermore, by performing more computations per parameter, PonderLMs achieve

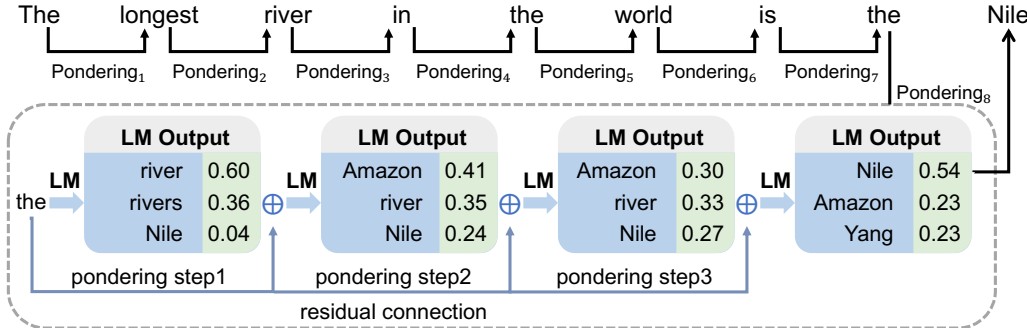

Figure 3: The inference process of a pondering language model, illustrated with an actual example from PonderPythia-2.8B, shows how the model dynamically corrects its predictions over several pondering steps to arrive at the correct answer. Crucially, the intermediate probability distributions and their top candidate tokens offer a potentially interpretable view into the model's inference.

higher parameter knowledge density (Allen-Zhu & Li, 2024), potentially reducing communication costs at scale.

Experimentally, by introducing the pondering process during pretraining, our Ponder GPT-2, LLaMA, and Pythia models achieve pretraining perplexity comparable to that of vanilla models with twice as many parameters. Furthermore, PonderPythia models significantly outperform the official Pythia models across nine popular downstream tasks, PonderPythia-2.8B surpasses Pythia-6.9B, and PonderPythia-1B is comparable to TinyLlama-1.1B, which is trained on 10 times more data. Notably, increasing the number of pondering steps consistently enhances model performance, underscoring the substantial potential of this approach.

Moreover, our method is orthogonal to traditional scaling strategies, including parameter scaling and inference-time scaling via CoT, and thus can complement existing techniques, potentially introducing a third scaling axis to enhance model performance.

## 2 PONDERING LANGUAGE MODEL

In this section, we introduce our proposed PonderLM (Figure 2), which integrates a pondering mechanism into language models via pretraining. Given that pretraining fundamentally constitutes a language modeling task, we briefly review this task before detailing our proposed model.

**Language Modeling.** Given a sequence of tokens $X = [x_1, x_2, \ldots, x_n]$, the primary objective of language modeling is typically to maximize the likelihood of predicting each token based on its preceding tokens. Formally, this is expressed through the joint probability factorization:

$$P(x_1, x_2, \ldots, x_n) = \prod_{t=1}^{n} P(x_t \mid x_{<t}) \tag{1}$$

Current language models first map tokens to input embeddings $\mathbf{E}^0 = [\mathbf{e}_1^0, \mathbf{e}_2^0, \ldots, \mathbf{e}_n^0]$, where each embedding $\mathbf{e}_i^0 \in \mathbb{R}^d$ is selected from a vocabulary embedding matrix $\mathbf{V} = [\mathbf{e}_1, \mathbf{e}_2, \ldots, \mathbf{e}_{|V|}]$, with vocabulary size $|V|$ and hidden dimension $d$. The language model then generates output probabilities $\mathbf{P}$ for predicting the next token at each position:

$$\mathbf{P} = \text{LM}(\mathbf{E}^0), \quad \mathbf{P} \in \mathbb{R}^{n \times |V|} \tag{2}$$

The cross-entropy loss is computed directly from these predicted probabilities $\mathbf{P}$ to pretrain the language model.

**Pondering Mechanism.** In our proposed method, instead of directly using the predicted output probabilities $\mathbf{P}$ to compute the cross-entropy loss, we utilize these probabilities as weights to sum embeddings of all candidate tokens, forming what we call a "pondering embedding". Given the

probability distribution $\mathbf{p} \in \mathbb{R}^{|V|}$ at each position, the pondering embedding $\mathbf{t}$ is:

$$\mathbf{t} = \sum_{i=1}^{|V|} p_i \mathbf{e}_i, \quad p_i \in \mathbb{R}, \mathbf{e}_i \in \mathbb{R}^d \tag{3}$$

For computational efficiency, pondering embeddings $\mathbf{t}$ for all positions can be calculated simultaneously via matrix multiplication[2]:

$$\mathbf{T} = \mathbf{PV}, \quad \mathbf{T} \in \mathbb{R}^{n \times d} \tag{4}$$

Through these pondering embeddings, we effectively map predicted probabilities back into the embedding space, preserving embedding information from all possible candidate tokens. To maintain the information from the original input embeddings, we integrate the pondering embeddings using a residual connection:

$$\mathbf{E}^1 = \mathbf{E}^0 + \mathbf{T} = [\mathbf{e}_1^0 + \mathbf{t}_1, \mathbf{e}_2^0 + \mathbf{t}_2, \ldots, \mathbf{e}_n^0 + \mathbf{t}_n] \tag{5}$$

We then feed the updated embeddings $\mathbf{E}^1$ back into the same language model to obtain refined output probabilities:

$$\mathbf{P}^1 = \mathrm{LM}(\mathbf{E}^1), \quad \mathbf{P}^1 \in \mathbb{R}^{n \times |V|} \tag{6}$$

After obtaining $\mathbf{P}^1$, we can iteratively repeat the previous process to achieve multi-step pondering. Specifically, given a predefined number of pondering steps $s$[3], we iteratively compute new pondering embeddings and integrate them with the original input embeddings using residual connections, feeding the result back into the same language model until $s$ steps are reached:

$$
\begin{aligned}
\mathbf{E}^0 &= [\mathbf{e}_1^0, \mathbf{e}_2^0, \ldots, \mathbf{e}_n^0], \quad \mathbf{P}^0 = \mathrm{LM}(\mathbf{E}^0), \quad \mathbf{T}^1 = \mathbf{P}^0 \mathbf{V} \\
\mathbf{E}^1 &= \mathbf{E}^0 + \mathbf{T}^1 = [\mathbf{e}_1^0 + \mathbf{t}_1^1, \mathbf{e}_2^0 + \mathbf{t}_2^1, \ldots, \mathbf{e}_n^0 + \mathbf{t}_n^1], \quad \mathbf{P}^1 = \mathrm{LM}(\mathbf{E}^1), \quad \mathbf{T}^2 = \mathbf{P}^1 \mathbf{V} \\
&\cdots \\
\mathbf{E}^s &= \mathbf{E}^0 + \sum_{i=1}^{s} \mathbf{T}^i = [\mathbf{e}_1^0 + \mathbf{t}_1^1 + \cdots + \mathbf{t}_1^s, \ldots, \mathbf{e}_n^0 + \mathbf{t}_n^1 + \cdots + \mathbf{t}_n^s], \quad \mathbf{P}^s = \mathrm{LM}(\mathbf{E}^s)
\end{aligned}
\tag{7}
$$

This iterative pondering mechanism progressively refines the model's predictions. Finally, we can use the refined output probabilities $\mathbf{P}^s$ after $s$ pondering steps to compute the cross-entropy loss and optimize the language model to perform $s$-step pondering.

## 3 EXPERIMENTS

Our experiments consist of 3 parts. First, we validate the scaling curves of pondering models on widely used GPT-2 and LLaMA architectures. Second, we perform large-scale pretraining of PonderPythia models on the Pile dataset and compare their scaling curves and language modeling capabilities with those of the official Pythia suite (Biderman et al., 2023). Third, we evaluate the downstream task performance of PonderPythia models, including 9 popular general tasks and an instruction-following task, and compare the results with official Pythia, OPT (Zhang et al., 2022), Bloom (Le Scao et al., 2023) and TinyLLaMA (Zhang et al., 2024).

### 3.1 SMALL SCALE VALIDATION ON GPT-2 AND LLAMA

We apply our proposed method to two popular Transformer architectures, GPT-2 and LLaMA, to investigate its general applicability and effectiveness.

**Experimental Settings.** We trained both models from 405M to 1.4B parameters from scratch on a subset of the Pile dataset using the same tokenizer. The amount of training data for each model follows the Chinchilla (Hoffmann et al., 2022b) scaling laws, with a fixed context length of 2048. Detailed configurations are specified in Appendix B.

---

[2]In practice, we use only the top-K tokens with highest probabilities at each position to compute the pondering embedding, reducing complexity from $\mathcal{O}(n|V|d)$ to $\mathcal{O}(nKd)$. With $K = 100 \ll |V|$, this does not degrade LM performance and makes the matrix multiplication overhead negligible within the overall LM computations. We have done the ablation study in Section 4.4.

[3]Unless otherwise specified, we set $s = 3$ for subsequent experiments.

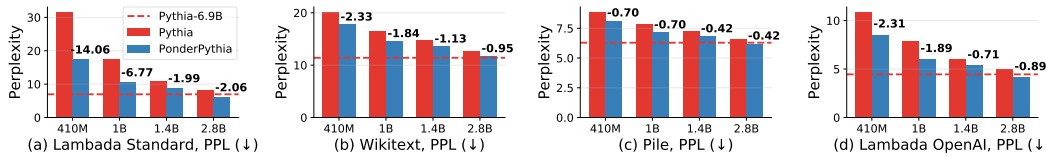

Figure 5: PonderPythia demonstrates substantial perplexity improvements over the official Pythia across all model sizes and datasets. Our PonderPythia-2.8B even surpasses the official Pythia-6.9B.

**Results.** As shown in Figure 4, the pondering mechanism significantly improves the performance of GPT-2 and LLaMA models across the 405M to 1.4B parameter range. For instance, an 834M pondering model achieves a loss comparable to a vanilla model trained with over $2\times$ the parameter-token product.

## 3.2 LARGE-SCALE PRETRAINING ON PILE

We further validate the effectiveness of our pondering method by conducting large-scale pretraining experiments on the entire Pile dataset (300B tokens) (Gao et al., 2020). We train a new model, named PonderPythia, from scratch using exactly the same architectural components (parallel attention and MLP layers, rotary embeddings with 1/4 head dimensions), same tokenizer and training hyperparameters (optimizer settings, learning rate schedule, batch size, and context length) as the original Pythia models. We then compare PonderPythia models' scaling curves and language modeling capabilities with the official Pythia model suite.

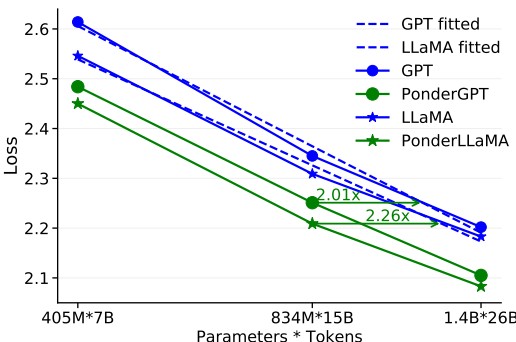

Figure 4: Scaling curves of GPT3 LLaMA and their corresponding pondering models.

## 3.2.1 SCALING CURVES

We plot the scaling curves of the PonderPythia and official Pythia models with respect to parameter size, training tokens, and training FLOPs. As shown in Figure 1 (left), the fitted curves indicate that a 2.55B-parameter PonderPythia model achieves a validation loss comparable to that of the 6.9B-parameter official Pythia model, while requiring 63% fewer parameters. In Figure 1 (middle), the fitted curves show that PonderPythia achieves performance comparable to the official Pythia model while using 59% fewer training tokens. In Figure 1 (right), we report the language modeling loss of vanilla Pythia-70M[4] and PonderPythia-70M under the same computational budget during pretraining. It can be observed that PonderPythia-70M consistently outperforms vanilla Pythia-70M when trained with the same number of FLOPs.

## 3.2.2 LANGUAGE MODELING ABILITY EVALUATION

We measure the perplexity on several language modeling datasets to reflect general language modeling capabilities. Specifically, we report perplexity scores on the Pile validation set, Wikitext, and the Lambda dataset (both OpenAI and standard versions), detailed in Figure 5. The results demonstrate significant perplexity improvements across all datasets and model sizes. Notably, the perplexity achieved by the PonderPythia-2.8B model is even better than that of the official Pythia-6.9B model.

## 3.3 DOWNSTREAM TASKS EVALUATION

We use previously pretrained PonderPythia models for downstream task evaluations.

---

[4]To match the training FLOPs of vanilla Pythia-70M with PonderPythia-70M, we trained the vanilla model for 4 epochs.

Table 1: Five-shot and zero-shot evaluations on downstream NLP tasks. All pretrained model weights used for comparison are obtained from their official repositories. Δacc is compared to the official Pythia models. Models in *italics* are excluded from bolding, as they use significantly larger training data or parameters.

| Model (#training tokens) | Lambada OpenAI | ARC -E | Lambada Standard | ARC -C | Wino Grande | PIQA | Hella Swag | SciQ | RACE | Avg acc / Δacc ↑ |
|---|---|---|---|---|---|---|---|---|---|---|
| *5-shot* | | | | | | | | | | |
| Pythia-410M (300B) | 43.9 | 54.7 | 32.8 | 22.3 | 53.4 | 68.0 | 33.8 | 88.9 | 30.4 | 47.6 |
| OPT-350M (300B) | 38.3 | 45.4 | 32.1 | 20.5 | 53.0 | 65.8 | 31.9 | 85.7 | 29.5 | 44.7 |
| Bloom-560M (366B) | 29.4 | 50.2 | 29.7 | 21.9 | 52.7 | 64.2 | 31.4 | 88.0 | 30.0 | 44.2 |
| **PonderPythia-410M** (300B) | **48.9** | **58.7** | **43.7** | **26.1** | **54.0** | **70.5** | **37.3** | **91.0** | **32.4** | **51.4** /+3.8 |
| Pythia-1B (300B) | 48.3 | 58.6 | 35.8 | 25.4 | 52.8 | 71.3 | 37.7 | 91.6 | 31.7 | 50.4 |
| OPT-1.3B (300B) | 54.0 | 60.4 | 49.0 | 26.9 | 56.9 | 72.4 | 38.5 | 91.8 | 35.4 | 52.7 |
| Bloom-1.1B (366B) | 36.3 | 54.9 | 37.4 | 24.9 | 53.4 | 67.6 | 34.8 | 88.7 | 33.0 | 47.9 |
| *Tinyllama-1.1B* (3T) | *53.8* | *64.8* | *45.0* | *31.1* | *59.4* | *73.8* | *44.9* | *94.0* | *36.4* | *55.9* |
| **PonderPythia-1B** (300B) | **57.7** | **63.2** | **52.5** | **28.6** | **58.6** | **73.3** | **41.9** | **93.4** | **36.3** | **56.2** / +5.8 |
| Pythia-1.4B (300B) | 54.5 | 63.1 | 44.5 | 28.8 | 57.1 | 71.0 | 40.5 | 92.4 | 34.6 | 54.1 |
| Bloom-1.7B (366B) | 42.5 | 58.8 | 41.5 | 26.2 | 57.7 | 68.7 | 37.6 | 91.9 | 33.5 | 50.9 |
| **PonderPythia-1.4B** (300B) | **59.2** | **67.5** | **49.9** | **32.4** | **60.4** | **73.5** | **44.2** | **94.3** | **37.1** | **57.6** / +3.5 |
| Pythia-2.8B (300B) | 59.0 | 67.0 | 50.7 | 31.0 | 61.1 | 74.4 | 45.3 | 93.7 | 35.9 | 57.6 |
| OPT-2.7B (300B) | 60.2 | 64.7 | 55.0 | 29.8 | 62.2 | 75.1 | 46.1 | 93.0 | 37.5 | 58.2 |
| Bloom-3B (366B) | 46.2 | 63.8 | 47.1 | 31.7 | 57.8 | 70.8 | 41.4 | 93.4 | 34.6 | 54.1 |
| *Pythia-6.9B* (300B) | *62.5* | *69.6* | *54.8* | *35.6* | *62.9* | *76.6* | *48.0* | *94.6* | *36.7* | *60.1* |
| *Pythia-12B* (300B) | *66.5* | *71.0* | *57.1* | *36.0* | *64.8* | *76.5* | *50.7* | *94.9* | *37.4* | *61.7* |
| **PonderPythia-2.8B** (300B) | **64.2** | **70.6** | **58.7** | **35.8** | **65.3** | **76.7** | **49.0** | **94.3** | **39.0** | **61.5** / +3.9 |
| *0-shot* | | | | | | | | | | |
| Pythia-410M (300B) | 51.4 | **52.2** | 36.4 | 21.4 | 53.8 | 66.9 | 33.7 | **81.5** | 30.9 | 47.6 |
| OPT-350M (300B) | 45.2 | 44.0 | 35.8 | 20.7 | 52.3 | 64.5 | 32.0 | 74.9 | 29.8 | 44.4 |
| Bloom-560M (366B) | 34.3 | 47.5 | 33.3 | 22.4 | 51.5 | 63.8 | 31.5 | 80.3 | 30.5 | 43.9 |
| **PonderPythia-410M** (300B) | **56.9** | 51.9 | **45.3** | **22.6** | **56.0** | **68.7** | **37.0** | 81.4 | **33.8** | **50.4** /+2.8 |
| Pythia-1B (300B) | 55.9 | 56.8 | 42.0 | 24.2 | 52.5 | 70.5 | 37.7 | 83.3 | 32.7 | 50.6 |
| OPT-1.3B (300B) | 57.9 | 57.1 | **52.5** | 23.4 | **59.7** | 71.8 | 41.6 | 84.3 | 34.3 | 53.6 |
| Bloom-1.1B (366B) | 42.6 | 51.5 | 42.9 | 23.6 | 54.9 | 67.3 | 34.5 | 83.6 | 32.6 | 48.2 |
| *Tinyllama-1.1B* (3T) | *58.8* | *60.3* | *49.3* | *28.0* | *59.0* | *73.3* | *45.0* | *88.9* | *36.4* | *55.4* |
| **PonderPythia-1B** (300B) | **62.3** | **60.5** | 51.9 | **27.0** | 56.5 | **72.2** | 41.8 | **87.4** | **35.4** | **55.0** / +4.4 |
| Pythia-1.4B (300B) | 61.6 | 60.4 | 49.7 | 25.9 | 57.5 | 70.8 | 40.4 | 86.4 | 34.1 | 54.1 |
| Bloom-1.7B (366B) | 46.2 | 56.4 | 44.5 | 23.7 | 56.8 | 68.5 | 37.5 | 85.0 | 33.2 | 50.2 |
| **PonderPythia-1.4B** (300B) | **65.2** | **62.0** | **53.8** | **27.0** | **60.1** | **72.6** | **44.0** | **89.0** | **35.2** | **56.5** / +2.4 |
| Pythia-2.8B (300B) | 64.6 | 64.4 | 54.3 | 29.5 | 60.2 | 73.8 | 45.4 | 88.5 | 34.9 | 57.3 |
| OPT-2.7B (300B) | 63.5 | 60.8 | 56.0 | 26.8 | 61.2 | 73.8 | 45.9 | 85.8 | 36.2 | 56.7 |
| Bloom-3B (366B) | 51.7 | 59.4 | 50.9 | 28.0 | 58.7 | 70.8 | 41.4 | 88.8 | 35.2 | 53.9 |
| *Pythia-6.9B* (300B) | *67.2* | *67.3* | *55.9* | *31.4* | *61.0* | *75.2* | *48.1* | *89.3* | *36.9* | *59.1* |
| *Pythia-12B* (300B) | *70.4* | *70.6* | *58.9* | *31.7* | *61.0* | *75.2* | *50.4* | *89.3* | *36.9* | *60.5* |
| **PonderPythia-2.8B** (300B) | **68.9** | **66.5** | **60.8** | **32.5** | **63.6** | **75.0** | **48.6** | **91.0** | **36.5** | **60.4** / +3.1 |

### 3.3.1 GENERAL DOWNSTREAM TASKS

We consider various widely-used benchmarks, including the tasks originally used by Pythia (LAM-BADA (Paperno et al., 2016), PIQA (Bisk et al., 2020), WinoGrande (Sakaguchi et al., 2021), ARC-E and ARC-C (Clark et al., 2018), SciQ (Welbl et al., 2017). We also include HellaSwag (Zellers et al., 2019) for commonsense reasoning and RACE (Lai et al., 2017) for reading comprehension.

We evaluate both 0-shot and 5-shot learning performance using the LM evaluation harness (Gao et al., 2023). Detailed results are shown in Table 1. Across all evaluated model sizes, PonderPythia consistently and significantly outperforms the official Pythia models, as well as comparable OPT and Bloom models. Remarkably, with only 1/10 of the training data (300B tokens) and fewer parameters,

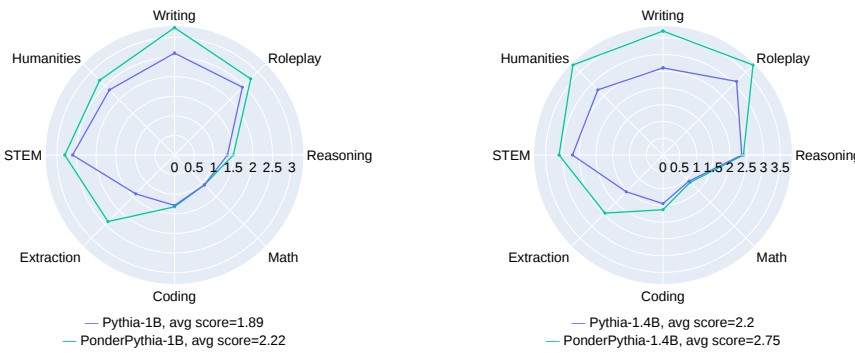

Figure 6: Instruction-following abilities evaluated on MT-Bench. PonderPythia-1B and 1.4B consistently outperform their corresponding official Pythia models across all subtasks.

our PonderPythia-1B achieves results comparable to, or even surpassing, TinyLlama-1.1B—which uses a more advanced LLaMA architecture and 3T tokens. Furthermore, PonderPythia-2.8B not only surpasses Pythia-6.9B but also achieves performance nearly on par with Pythia-12B.

### 3.3.2 INSTRUCTION-FOLLOWING ABILITY EVALUATION

To assess the instruction-following capability of our model, we further fine-tuned PonderPythia-1B and 1.4B, as well as the corresponding official Pythia models, on the Alpaca dataset using the same settings (Taori et al., 2023). The fine-tuned models were evaluated with MT-Bench (Zheng et al., 2023), a popular multi-turn question benchmark. The experimental results are shown in Figure 6. As illustrated, both PonderPythia-1B and 1.4B consistently outperform their official Pythia counterparts across all subtasks, achieving average improvements of 0.33 and 0.55, respectively.[5]

## 4 ABLATION STUDY

In this section, we conduct a series of ablation studies to dissect the components of our proposed pondering language model. We use a 70M parameter Pythia model trained on a 30B-token subset of the Pile dataset as our primary testbed to ensure controlled and efficient experimentation.

### 4.1 COMPARISON WITH RELATED BASELINES

First, we compare our pondering mechanism against several baselines designed to increase computation per token. The goal is to verify that the performance gains come from our specific approach rather than merely from additional computation. The baselines include:

- **Last Hidden State as Embedding**: Replacing the pondering embedding with the last hidden state from the previous step, similar to Coconut (Hao et al., 2024).
- **Projected Hidden State**: A variant of the above where a linear projector (like LLaVA (Liu et al., 2023)) maps the last hidden state to the embedding space.
- **Looped Transformer** (Giannou et al., 2023; Saunshi et al.): Increases computation by iteratively reusing the full stack of transformer layers.
- **Pause Token** (Goyal et al., 2023): A method that inserts a special learnable "pause" token after each original token to encourage extra computation.

As shown in Table 2, our method significantly outperforms all baselines across every benchmark given the same number of additional steps/loops/pauses. Notably, our model with only 3 pondering steps is more effective than baselines with 6 loops, steps or pauses, confirming that the performance gains are attributable to our specific approach rather than just added computation.

---

[5]The marginal gains on Coding and Math tasks may be attributed to the limited coding and mathematical abilities of the Pythia models.

Table 2: Comparison on various benchmarks. PonderPythia-70M consistently outperforms all baseline methods, with the 3-step version already surpassing 6-step/loop/pauses baselines.

| Model | Pile(↓) | Wikitext(↓) | Lambada OpenAI(↓) | Lambada Standard(↓) | Avg Acc 0 shot(↑) | Avg Acc 5 shot(↑) |
|---|---|---|---|---|---|---|
| Pythia-70M (baseline) | 16.95 | 51.68 | 101.34 | 481.04 | 21.04 | 14.12 |
| *Models with 3 additional steps/loops/pauses* | | | | | | |
| Looped Pythia-70M (3 loops) | 15.33 | 44.56 | 74.75 | 438.62 | 22.67 | 17.21 |
| Pause Pythia-70M (3 pauses) | 16.53 | 49.85 | 80.77 | 374.97 | 21.62 | 15.67 |
| Last Hidden State (3 steps) | 15.64 | 45.60 | 78.11 | 479.85 | 22.11 | 15.44 |
| + Linear Projector (3 steps) | 15.64 | 46.10 | 81.94 | 453.78 | 21.62 | 17.69 |
| **PonderPythia (3 steps)** | **14.16** | **39.63** | **56.01** | **238.45** | **25.13** | **20.64** |
| *Models with 6 additional steps/loops/pauses* | | | | | | |
| Looped Pythia-70M (6 loops) | 15.18 | 43.91 | 71.78 | 355.64 | 22.50 | 17.44 |
| Pause Pythia-70M (6 pauses) | 16.55 | 49.66 | 80.79 | 460.80 | 21.61 | 15.88 |
| Last Hidden State (6 steps) | 15.30 | 44.48 | 78.87 | 393.03 | 22.46 | 18.18 |
| + Linear Projector (6 steps) | 15.29 | 44.24 | 74.36 | 474.15 | 21.69 | 17.61 |
| **PonderPythia (6 steps)** | **13.56** | **37.57** | **49.15** | **196.67** | **25.58** | **21.71** |

## 4.2 Impact of Pondering Steps

To further investigate the effect of pondering steps on model performance, we trained several 70M-parameter Pythia from scratch with different numbers of pondering steps: 0 (baseline), 1, 2, 3, 4, 5, and 10. The results in Figure 7 (top) show that increasing pondering steps consistently reduces the language modeling loss on the Pile validation set, demonstrating the potential of our method.

## 4.3 Training with Randomized Pondering Steps

In our main experiments, the number of pondering steps is fixed during training and inference due to source limit. To build a more flexible model, we also experimented with randomizing the number of pondering steps during training. We trained a Pythia-70M model where the number of steps for each training batch was randomly sampled from the range $[1, 10]$.

This strategy yields a single model that can operate with a variable number of steps at inference. As shown in Figure 7 (bottom), this model exhibits test-time scaling: its performance progressively improves as we increase the number of pondering steps, and thus the computational budget, at test time.

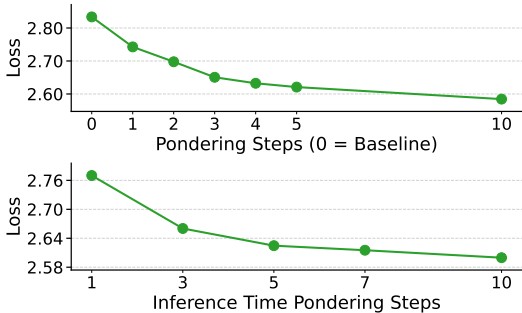

Figure 7: (Top) Increasing the number of pondering steps consistently reduces the validation loss. (Bottom) Inference-time scaling of a model trained with randomized pondering steps.

## 4.4 Influence of Top-K Token Selection

To balance performance and efficiency, we evaluated the effect of the hyperparameter $K$ on model perplexity. Increasing $K$ from 10 to 100 substantially reduced perplexity from 15.18 to 14.21. However, expanding the selection to the full vocabulary yielded only a negligible improvement to 14.20. This indicates that tokens with ranks between 10 and 100 contribute substantially, while those beyond offer negligible benefit. We thus adopt $K = 100$ in our main experiments.

## 5 Related Work

**Test-Time Compute Scaling.** Scaling computation at test time has proven effective for improving model performance without adding parameters (Snell et al., 2024), with existing methods mainly categorized into parallel and sequential scaling (Zeng et al., 2024; Muennighoff et al., 2025).

Table 3: A Comparison of PonderLM with Related Methods.

| Method | Training Data | Computation Space | Application Level | Training Method |
|---|---|---|---|---|
| CoT | CoT data | Explicit Token | Per question | RL/SFT |
| LoopedLM | General corpus | Hidden State | Per token | Pretrain |
| Pause Tokens | General corpus | Fixed Token | Per token | Pretrain |
| Quiet-STaR | General corpus | Explicit Token | Per token | RL |
| Coconut | CoT data | Hidden State | Per question | SFT |
| PonderLM | General corpus | Weight Sum of Embeddings | Per token | Pretrain |

*Parallel scaling* generates multiple candidates simultaneously and selects the best via strategies like Best-of-N (BoN) (Cobbe et al., 2021; Sun et al., 2024; Gui et al., 2024; Amini et al., 2024; Sessa et al., 2024) or Majority Voting (Wang et al., 2022), but faces the challenge of reliably identifying the optimal candidate (Stroebl et al., 2024; Hassid et al., 2024).

*Sequential scaling* methods improve reasoning by iteratively refining a model's output over multiple steps. This broad category includes foundational techniques like Chain of Thought (CoT) as well as more recent iterative revision strategies (Wei et al., 2022; Nye et al., 2021; Huang et al., 2022; Min et al., 2024; Madaan et al., 2024; Wang et al., 2024b; Lee et al., 2025; Hou et al., 2025; Muennighoff et al., 2025; Li et al., 2025). State-of-the-art models also scale computation at test-time through extensive multi-step reasoning (OpenAI, 2024; DeepSeek-AI et al., 2025; Comanici et al., 2025). However, these methods often depend on specialized datasets (Allen-Zhu & Li, 2023), require long context windows (Zhu et al., 2025), or involve complex reinforcement learning (Pang et al., 2025). Our method is designed to overcome these limitations.

**Latent Thinking in Language Models.** Latent thinking refers to the intermediate computational processes that occur within a language model's internal representations, separate from the explicit generation of text tokens (Yang et al., 2024; Biran et al., 2024). Prior work can be broadly categorized by the computational space in which this additional thinking occurs.

*Discrete Token Space.* One line of research elicits intermediate reasoning by manipulating discrete tokens. This includes predicting dedicated planning tokens (Wang et al., 2024a), using filler tokens to allocate more computation (Pfau et al., 2024), or inserting learnable "pause" tokens during training to encourage latent computation (Zhou et al., 2024; Goyal et al., 2023). More complex methods like Quiet-STaR use reinforcement learning to generate and then condense rationales at each token, embedding the reasoning process directly into generation (Zelikman et al., 2024).

*Continuous Space.* Another research direction explores reasoning directly within the model's continuous hidden states. One prominent approach involves iteratively refining these states. This is achieved through recurrent structures that reuse hidden states over time (Dehghani et al.; Hutchins et al., 2022), by recycling model outputs back as inputs, a technique effective for reasoning tasks (Giannou et al., 2023; Yang et al., 2023; Saunshi et al.), or by iterating over model layers to refine intermediate representations (Geiping et al., 2025). More recently, methods like Coconut (Hao et al., 2024) train models to reason entirely in the latent space, while CoCoMix (Tack et al., 2025) extract the "concept" from the hidden states to improve language modeling.

In contrast to these approaches that operate on hidden states, our method operates on pondering embeddings derived from the model's predictive probability distributions. We further compare our method with the most relevant prior work in Table 3.

## 6 CONCLUSION

In this paper, we introduce the pondering process into language models through solely self-supervised learning. PonderLM can be naturally learned through pretraining on large-scale general corpora. Our extensive experiments across three widely adopted architectures—GPT-2, Pythia, and LLaMA—highlight the effectiveness and generality of our proposed method. Notably, our PonderPythia consistently outperforms the official Pythia model on language modeling tasks, scaling curves, downstream tasks, and instruction-following abilities when pretrained on the large-scale Pile dataset. As increasing the number of pondering steps further improves language model performance, we posit that our approach introduces a promising new dimension along which language model capabilities can be scaled.

## 7 ACKNOWLEDGEMENTS

This work is sponsored by the National Natural Science Foundation of China (NSFC) grant (No. 62576211) and the National Key Research and Development Program of China (No. 2023ZD0121402).

## 8 ETHICS STATEMENT

This paper introduces a novel pre-training architecture for language models that achieves significant performance improvements at a given parameter scale. However, we acknowledge that, like other powerful language models, this technology is susceptible to misuse for malicious or illegal purposes. Furthermore, as a new architecture, our proposed method may contain unforeseen security vulnerabilities, posing potential risks to data privacy and system integrity.

## 9 REPRODUCIBILITY STATEMENT

To facilitate reproducibility, the source code for our core models and experiments is available in the supplementary material. We have documented all essential hyperparameters and implementation details in the appendix. We believe this provides sufficient information for the research community to verify and reproduce the results presented in this work.

## 10 THE USE OF LARGE LANGUAGE MODELS

The core methodology and conceptual framework presented in this paper were developed without the assistance of Large Language Models (LLMs). The use of LLMs was strictly limited to refining grammatical structure and enhancing the academic expression of the text.

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

## A    TRAINING DETAILS OF THE MAIN RESULT

The main computational cost was incurred during the pretraining of PonderPythia-1B, 1.4B, and 2.8B on the 300B-token Pile dataset. We pretrained these models on a cluster of high-performance GPUs with 64GB memory, which required 19,886, 31,680, and 109,570 GPU hours, respectively.

## B    TRAINING DETAILS OF THE SCALING LAW

In Section 3.1, we have discussed the basic scaling law of our pondering models. These hyperparameters primarily follow the GPT-3 specifications (Brown et al., 2020). However, unlike GPT-3, we untie the input and output embedding matrices. We specify the parameters of our models and the training hyperparameters in Table 4.

Table 4: Model sizes and hyperparameters for scaling experiments.

| params | $n_{layers}$ | $d_{model}$ | $n_{heads}$ | learning rate | batch size (in tokens) | tokens |
|---|---|---|---|---|---|---|
| 405M | 24 | 1024 | 16 | 3e-4 | 0.5M | 7B |
| 834M | 24 | 1536 | 24 | 2.5e-4 | 0.5M | 15B |
| 1.4B | 24 | 2048 | 32 | 2e-4 | 0.5M | 26B |

## C    LIMITATIONS AND FUTURE WORK

### C.1    LIMITATIONS

There are two limitations to our work. Firstly, due to computational constraints, we only scaled our method up to a 2.8B-parameter model trained on 300B tokens from the Pile dataset. It would be interesting to extend our approach to larger models and larger-scale datasets in the future. Secondly, although our results demonstrate that the proposed method scales better than vanilla models under the same training FLOPs (Figure 1), it also introduces additional inference overhead (increasing roughly linearly with the number of pondering steps), similar to test-time scaling methods.

### C.2    FUTURE WORK

There are several promising directions for future work. Firstly, our proposed method is not limited to decoder-only architectures or language modeling; it has the potential to be applied to a wide range of model types and domains. For example, it could be adapted to state-space models such as Mamba (Gu & Dao, 2023), encoder models, or RWKV (Peng et al., 2023), as well as extended to other areas. Another promising direction is the introduction of token-adaptive pondering, which may significantly reduce computation and further enhance our method. It would also be interesting to investigate the interpretability of the pondering process, such as how the model "thinks" during pondering, the semantics of the pondering embedding, and whether the model learns to reflect on its predictions through pondering. Finally, exploring the combination of our method with orthogonal approaches such as CoT and other test-time scaling methods could also be an interesting direction.

## D    INTEGRATING PONDERING VIA CONTINUAL PRETRAINING

To investigate if our pondering mechanism can be integrated into existing large language models, we conducted a continual pretraining experiment. We started with the pre-trained Pythia-1B model and continued its training on a 30-billion-token subset of The Pile. We compare two approaches: a standard continual pretraining of the vanilla model (Pythia-1B-Vanilla-CPT) and continual pretraining with our pondering mechanism (PonderPythia-1B-CPT).

Our results show that this integration is effective. As seen in the training loss curves in Figure 8, PonderPythia consistently achieves a lower loss, demonstrating superior learning efficiency. This

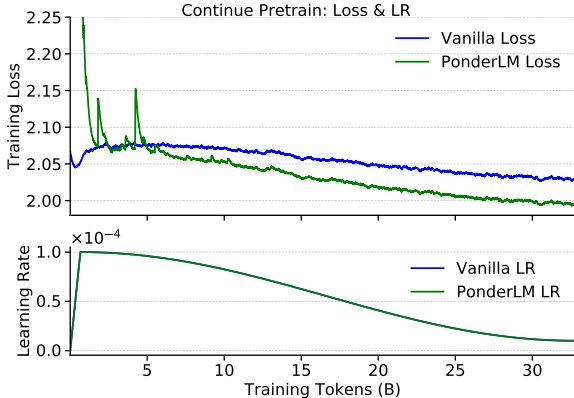

Figure 8: Training loss during continual pretraining. PonderPythia-1B achieves a lower loss than the vanilla baseline, suggesting greater learning efficiency.

efficiency translates to improved downstream performance, as detailed in Table 5. The Ponder-Pythia model achieves a higher average accuracy in both 5-shot and 0-shot settings across a suite of benchmarks.

Collectively, these findings confirm that continual pretraining is a viable and effective strategy for equipping pre-trained models with our pondering mechanism, enhancing both adaptation efficiency and downstream capabilities.

Table 5: Downstream task performance after continual pretraining on 30B tokens. PonderPythia-1B shows improved average accuracy in both 5-shot and 0-shot evaluations.

| Model (#training tokens) | Lambada OpenAI | ARC -E | Lambada Standard | ARC -C | Wino Grande | PIQA | Hella Swag | SciQ | RACE | Avg acc / $\Delta$acc ↑ |
|---|---|---|---|---|---|---|---|---|---|---|
| | | | | *5-shot* | | | | | | |
| Pythia-1B-Vanilla-CPT | 49.4 | 60.0 | 36.5 | 27.1 | 51.9 | 71.4 | 37.9 | 90.4 | 31.9 | 50.7 |
| PonderPythia-1B-CPT | 49.5 | 58.8 | 42.6 | 25.4 | 54.3 | 69.2 | 37.9 | 91.2 | 34.7 | 51.5 (+0.8) |
| | | | | *0-shot* | | | | | | |
| Pythia-1B-Vanilla-CPT | 56.7 | 56.3 | 42.7 | 24.6 | 52.5 | 70.6 | 37.7 | 83.6 | 32.6 | 50.8 |
| PonderPythia-1B-CPT | 56.7 | 56.3 | 45.4 | 23.8 | 55.6 | 68.3 | 37.8 | 86.3 | 33.0 | 51.5 (+0.7) |

# E    SCALING LAWS WITH DIFFERENT PONDERING STEPS

To further validate the robustness of our method, we extended our analysis to LLaMA models trained with different numbers of pondering steps. Specifically, we tested configurations with 2 and 4 steps while keeping all other experimental settings constant. As illustrated in Figure 9, our method remains consistently effective, with model performance improving further as the number of pondering steps increases.

# F    ANALYSIS ON INPUT EMBEDDINGS AND OUTPUT DISTRIBUTIONS

To better understand the dynamics of the pondering process, we analyze the properties of input embeddings and output distributions using the Pythia-70M model trained with 10 pondering steps (as described in Section 4.2). We conduct the evaluation on a batch of 64 validation sequences with a context length of 2048 tokens.

## F.1    INPUT EMBEDDINGS

We first examine the evolution of the input embeddings $E^s$ across pondering steps.

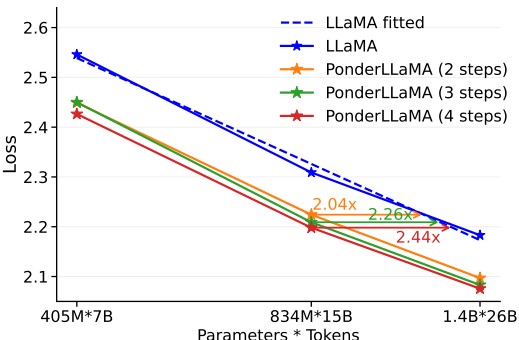

Figure 9: Scaling curves of LLaMA with different pondering steps.

**Cosine Similarity.** To measure the convergence of the embeddings, we calculate the cosine similarity between consecutive steps, $Cosine(E^{s-1}, E^s)$. Here we define $s = 0$ as the original input embedding (before pondering). As shown in Figure 10, the similarity starts at approximately 0.88 and gradually converges to 1.0, indicating that the input representations stabilize over time.

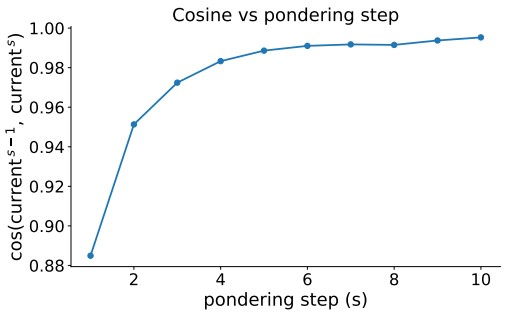
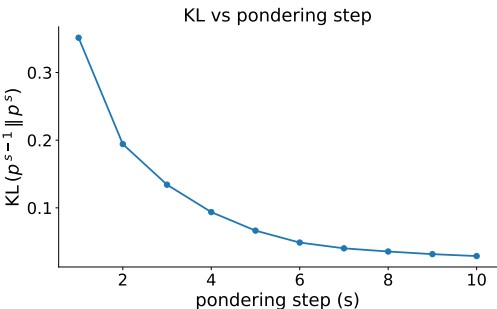

Figure 10: The cosine similarity between consecutive embedding states $E^{s-1}$ and $E^s$ across pondering steps.

Figure 11: KL divergence between consecutive output distributions, averaged over tokens.

**Spectral Properties.** We further analyze the geometry of the embeddings. Figure 12 visualizes the spectral energy distribution (explained variance ratio) of the top singular values. Consistent with this distribution, we observe a steady decrease in Effective Rank and a simultaneous increase in Cumulative Variance (Figure 12). This collective evidence indicates that the embedding energy progressively concentrates into dominant components during the pondering process.

### F.2 OUTPUT DISTRIBUTIONS

We also monitor the changes in the model's predictions by calculating the Kullback-Leibler (KL) divergence between the output probability distributions $P^s$ of consecutive steps, $D_{KL}(P^{s-1}\|P^s)$.

As illustrated in Figure 11, the KL divergence exhibits large variations in the initial steps (starting around 0.35) and decreases monotonically thereafter, suggesting that the model's predictions refine quickly and then settle into a stable state.

## G CASE STUDIES

To provide a deeper insight into the internal pondering process of PonderLM, we present case studies derived from the pre-trained PonderPythia-2.8B model. The following tables visualize the evolution of the top-3 candidate tokens and their corresponding probabilities across sequential pondering steps.

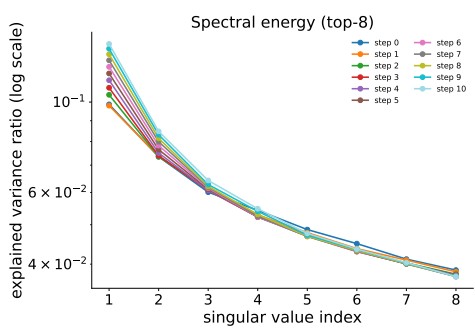 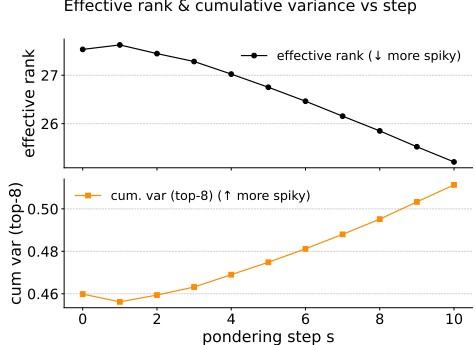

Figure 12: Left: Spectral energy (explained variance ratio, log scale) of the top-8 singular values of the token-embedding matrix at each pondering step $s$. Right: Effective rank ($\downarrow$ more spiky) and cumulative variance of the top-8 components ($\uparrow$ more spiky) versus step $s$, indicating progressive concentration of energy into few dominant directions.

The speed of light in vacuum is approximately 3 times ten to the power of

| Output Probs | Pondering steps 1 | Pondering steps 2 | Pondering steps 3 | Final predicted |
|---|---|---|---|---|
| Rank 1 | logarithm (0.45) | ˆ - (0.38) | _8 (0.48) | **8** (0.55) |
| Rank 2 | , (0.38) | aggreg (0.32) | **8** (0.36) | eight (0.23) |
| Rank 3 | ten (0.18) | approximately (0.31) | tion (0.17) | 10 (0.22) |

The chemical symbol for silver is

| Output Probs | Pondering steps 1 | Pondering steps 2 | Pondering steps 3 | Final predicted |
|---|---|---|---|---|
| Rank 1 | symbol (0.47) | atoms (0.39) | **Ag** (0.40) | **Ag** (0.94) |
| Rank 2 | symbols (0.32) | atomic (0.32) | symbol (0.33) | ” (0.04) |
| Rank 3 | nickname (0.20) | elemental (0.30) | elements (0.27) | S (0.02) |

The largest ocean on Earth is the

| Output Probs | Pondering steps 1 | Pondering steps 2 | Pondering steps 3 | Final predicted |
|---|---|---|---|---|
| Rank 1 | oceans (0.66) | **Pacific** (0.40) | **Pacific** (0.59) | **Pacific** (0.77) |
| Rank 2 | ocean (0.32) | oceans (0.32) | oceans (0.21) | Atlantic (0.19) |
| Rank 3 | seas (0.02) | Antarctic (0.28) | ocean (0.20) | Indian (0.03) |

The derivative of sin x is

| Output Probs | Pondering steps 1 | Pondering steps 2 | Pondering steps 3 | Final predicted |
|---|---|---|---|---|
| Rank 1 | differentiable (0.55) | homework (0.61) | approximately (0.37) | **cos** (0.36) |
| Rank 2 | the (0.23) | differentiable (0.21) | **cos** (0.32) | x (0.34) |
| Rank 3 | derivatives (0.21) | basics (0.17) | differentiable (0.31) | (0.30) |

The opposite of north is

| Output Probs | Pondering steps 1 | Pondering steps 2 | Pondering steps 3 | Final predicted |
|---|---|---|---|---|
| Rank 1 | directions (0.41) | directions (0.40) | **south** (0.92) | **south** (0.95) |
| Rank 2 | compass (0.36) | noun (0.32) | unclear (0.04) | east (0.02) |
| Rank 3 | the (0.22) | idiot (0.28) | South (0.03) | not (0.02) |

The chemical symbol for gold is

| Output Probs | Pondering steps 1 | Pondering steps 2 | Pondering steps 3 | Final predicted |
|---|---|---|---|---|
| Rank 1 | symbol (0.48) | atoms (0.45) | ͟Au (0.78) | **Au** (0.87) |
| Rank 2 | symbols (0.40) | elemental (0.28) | **Au** (0.17) | ” (0.09) |
| Rank 3 | approximately (0.12) | metals (0.27) | element (0.05) | the (0.04) |

# H GRADIENT NORMS COMPARISON

In this section, we take a closer look at training stability by comparing the gradient norms of the PonderLM against the vanilla GPT-2 1.4B model (Section 3.1). Both models were trained from scratch on the Pile using the exact same data and hyperparameters. As shown in Figure 13, while the pondering model shows a few minor spikes early on, it quickly recovers and maintains a stable training trajectory similar to the baseline.

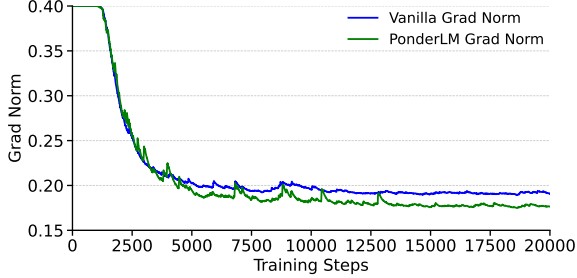

Figure 13: Gradient norms during pre-training for vanilla vs. PonderLM 1.4B.

