# OpenReview forum: "PonderLM: Pretraining Language Models to Ponder in Continuous Space"
_ICLR.cc/2026/Conference — ICLR 2026 Poster_

### Official Review · Reviewer_We2v · 2025-10-26

**Soundness:** 3
**Presentation:** 3
**Contribution:** 3
**Rating:** 6
**Confidence:** 4

**Summary:**

The paper introduces a novel technique (called "pondering") to iteratively call the forward pass of an LLM for a single token generation. Typically, a forward pass with an LLM over an input sequence results in a probability distribution over the vocabulary, which is then used for decoding the output token. The pondering approach in this paper utilizes the probability distribution at each token position to compute a weighted sum of the vocab embeddings. The main idea is to leverage such refined embeddings in subsequent forward passes ("s" times), followed by the calculation of the CE loss. The results show that the pondering approach improves the FLOPs and data usage efficiency during pre-training of various model architectures and sizes.

**Strengths:**

1. The methodology is well presented along with clear writing, code, and results.

2. The language modeling and downstream task evaluation results are promising and can influence research on such iterative embedding refinements as a new frontier to explore the scaling of compute.

**Weaknesses:**

1. A major concern I have is about the validity of the scaling law shown in Figure 4 if one were to change the number of pondering steps from 3 to a lower or higher value. Or even if the randomized pondering step approach was employed for that matter. If the scaling laws were similar to figure 4 with a randomized approach of pondering step selection, then that is an even better approach to scale since it is currently unclear how the value of (s=3 steps) was chosen. I believe the message of the paper can be further strengthened if this weakness is addressed.

2. This is a minor concern, but can a model trained to ponder for (s=3) steps be effective when pondering for (s>3) steps during inference? Some early results (either positive/negative) can influence a lot of future work in this direction.

**Questions:**

1. Did the authors analyze any properties of the embeddings after every pondering step? For ex: spectral properties, cosine similarities etc

2. How do the gradient norms change when incorporating pondering vs the vanilla pretraining?

3. Were there any training stability challenges when training from scratch, since I see some spikes in Figure 8 for continued pre-training? How did you address them?

4. An analysis of how the output distributions evolve with iterative pondering steps would be a great addition to the paper. For ex, does the KL divergence of the output distributions change drastically in early pondering steps or later?

---

> ### Author Response · Authors · 2025-11-25
> **Response to Reviewer We2v (Part 1/2)**
>
> Thank you for recognizing our clear methodology and the potential of our work to open a new frontier in compute scaling. Below, we provide point-by-point responses to your comments.
>
> > **[W1]** A major concern I have is about the validity of the scaling law shown in Figure 4 if one were to change the number of pondering steps from 3 to a lower or higher value. Or even if the randomized pondering step approach was employed for that matter. If the scaling laws were similar to figure 4 with a randomized approach of pondering step selection, then that is an even better approach to scale since it is currently unclear how the value of (s=3 steps) was chosen. I believe the message of the paper can be further strengthened if this weakness is addressed.
>
> We have addressed these points as follows:
> - Scaling Laws with Different Pondering Steps: To demonstrate the validity of our scaling laws across different configurations, we have added Appendix J, where we plot the scaling curves (**Appendix Figure 12**) for LLaMA models trained with 2 and 4 pondering steps (keeping all other settings constant). These results confirm that our method remains consistently effective, with model performance improving further as the number of pondering steps increases. We also present the results in the following:
>
>
> | Model\Pondering steps | s=0(vanilla) | s=2 | s=3|s=4|
> | -------- | -------- | -------- | -------- | -------- |
> |Llama-410M|	2.546|	2.449|	2.450|	2.426|
> |Llama-834M|	2.309|	2.224|	2.209|	2.198|
> |Llama-1.4B|	2.183|	2.097|	2.083|	2.076|
>
>
> - Randomized Pondering Steps: We agree that randomized steps offer significant advantages. As discussed in **Section 4.3** and shown in **Figure 7 (bottom)**, a model trained with randomized steps effectively exhibits test-time scaling, allowing performance to progressively improve by increasing steps at inference. We did not employ this strategy in our main large-scale experiments solely due to computational resource constraints.
> - Selection of s=3: The choice of s=3 was not arbitrary. Based on our ablation study in **Section 4.2 (Figure 7, top)**, we selected s=3 to strike a balance between performance gains and computational cost. However, our results explicitly show that increasing $s$ yields even lower loss, further demonstrating the potential of the PonderLM method.
>
>
>
> > **[W2]** This is a minor concern, but can a model trained to ponder for (s=3) steps be effective when pondering for (s>3) steps during inference? Some early results (either positive/negative) can influence a lot of future work in this direction.
>
> Thank you for this suggestion. If we fix the number of pondering steps(s=3)  during training but pondering more steps during inference, the model does not adapt well to this mismatch, and performance degrades. For example, the following table shows the validation PPL on pile of Pondering Pythia-1.4B when evaluated with different numbers of pondering steps (s):
>
> | inference pondering steps| s=3 | s=4|s=5|
> | -------- | -------- | -------- | -------- |
> |perplexity|	6.82| 7.52| 10.07|
>
> However, as noted in our previous response, this limitation can be addressed by the "randomized pondering steps" strategy (Section 4.3). By exposing the model to varying steps during training, it successfully learns to utilize additional compute at test time (Figure 7, bottom).
>
>
> > **[Q2]** How do the gradient norms change when incorporating pondering vs the vanilla pretraining?
>
> We plot the gradient norms for the 1.4B model trained from scratch on the Pile in **Appendix Figure 13**. Although we observed minor gradient spikes during the early-to-mid training stages when incorporating pondering, the model demonstrated strong resilience, quickly recovering stability.
>
>
> > **[Q3]** Were there any training stability challenges when training from scratch, since I see some spikes in Figure 8 for continued pre-training? How did you address them?
>
> Compared to vanilla transformer training, we did observe some instability, manifested as a series of spikes. However, we adopted several measures to stabilize training, including using a variance of $\sigma^2 = \frac{2}{5\times \text{hiddensize}}$ for initialization and scaling the output of the embedding layer by $\sqrt{\text{hiddensize}}$[1]. After these adjustments, training became more stable, and we were able to increase pondering steps to 10 (Section 4.2) and scale up the model size to 2.8B. We believe that more pondering steps and larger model sizes remain applicable.
>
> [1] Spike No More: Stabilizing the Pre-training of Large Language Models

---

> ### Author Response · Authors · 2025-11-25
> **Response to Reviewer We2v (Part 2/2)**
>
> > **[Q1&Q4]** Did the authors analyze any properties of the embeddings after every pondering step? For ex: spectral properties, cosine similarities etc
> > An analysis of how the output distributions evolve with iterative pondering steps would be a great addition to the paper. For ex, does the KL divergence of the output distributions change drastically in early pondering steps or later?
>
> Thank you for this insightful suggestion! We have added a new section in Appendix H to analyze the evolution of the input embeddings and output distributions. Specifically, we visualize the cosine similarity and KL divergence in **Appendix Figures 9 and 10**, and the spectral properties in **Appendix Figure 11**.
>
> Our observations are as follows:
> - **Input Embeddings**: The cosine similarity between consecutive embeddings (Appendix Figure 9) changes significantly at the start and then gradually converges to 1.0. Simultaneously, we observe a steady decrease in effective rank and an increase in the cumulative variance of the top components (Appendix Figure 11). This indicates that the embedding energy progressively concentrates during the pondering process. We also present the results in the following (s=0 means the original input embedding):
>
> | Pondering step | s=0 | s=1 | s=2 | s=3 | s=4 | s=5 | s=6 | s=7 | s=8 | s=9 | s=10 |
> | --- | --- | --- | --- | --- | --- | --- | --- | --- | --- | --- | --- |
> | Cos | - | 0.885 | 0.951 | 0.972 | 0.983 | 0.989 | 0.991 | 0.992 | 0.992 | 0.994 | 0.995 |
> | Effective rank | 27.533 | 27.626 | 27.446 | 27.283 | 27.023 | 26.751 | 26.463 | 26.155 | 25.850 | 25.523 | 25.210 |
> | Cumulative variance (top-8) | 0.496 | 0.456 | 0.459 | 0.463 | 0.469 | 0.475 | 0.481 | 0.488 | 0.495 | 0.503 | 0.511 |
>
>
> - **Output Distributions**: Similarly, the KL divergence (Appendix Figure 10) exhibits large variations in the initial steps and decreases monotonically thereafter. We also present the results in the following:
>
> | Pondering step | s=1 | s=2 | s=3 | s=4 | s=5 | s=6 | s=7 | s=8 | s=9 | s=10 |
> | --- | --- | --- | --- | --- | --- | --- | --- | --- | --- | --- |
> | KL divergence | 0.351 | 0.194 | 0.134 | 0.094 | 0.066 | 0.049 | 0.040 | 0.035 | 0.032 | 0.029 |

---

### Official Review · Reviewer_w4mh · 2025-10-28

**Soundness:** 2
**Presentation:** 3
**Contribution:** 2
**Rating:** 4
**Confidence:** 3

**Summary:**

The paper introduces PonderLM, a self-supervised, architecture-agnostic mechanism that lets a language model “ponder” within a single token step by feeding back a probability-weighted sum of token embeddings for additional forward passes, creating a continuous, fully differentiable inner loop that increases compute per parameter without RL or curated CoT data. Implemented across GPT-2, Pythia, and LLaMA, the approach yields lower pretraining perplexity and strong downstream gains—e.g., PonderPythia-2.8B surpasses Pythia-6.9B and rivals Pythia-12B, while PonderPythia-1B matches TinyLlama-1.1B trained on 10× more data—with performance improving as pondering steps increase.

The key contributions are: (i) a simple pondering loop that replaces discrete token emission with continuous weighted embeddings; (ii) proof that such behavior emerges via standard next-token pretraining alone; (iii) consistent benefits across model families and scales, especially for smaller models; and (iv) a framing of pondering as a third, orthogonal scaling axis (complementary to parameter and CoT test-time scaling) that may improve parameter knowledge density and reduce communication costs at scale.

**Strengths:**

1. The paper introduces a differentiable, self-supervised pondering loop that feeds a probability-weighted embedding back into the model within a single token generation step. This elegant mechanism eliminates the discrete bottleneck imposed by vocabulary spaces during internal computation. By conceptualizing pondering as a third scaling axis, orthogonal to both parameter scaling and test-time CoT scaling, the work offers a novel perspective on model scaling dynamics. Moreover, demonstrating that this behavior can emerge without reinforcement learning or curated CoT supervision substantially relaxes the conventional prerequisites of test-time scaling approaches.

2. The proposed inner loop is straightforward, easily integrable into standard language model architectures, and fully differentiable for end-to-end training using conventional next-token prediction. Empirical evaluations across three architectures (GPT-2, Pythia, and LLaMA) and nine downstream tasks reveal consistent and substantial performance gains—most notably, size-for-size improvements where a 2.8B model outperforms a 6.9B counterpart and approaches the performance of a 12B model. Moreover, the results exhibit monotonic improvements as the number of pondering steps increases.

**Weaknesses:**

1. The paper’s motivation and theoretical foundation appear insufficiently developed. It remains unclear why repeating the forward pass within a single token-generation step should improve performance. The current justification—an analogy to human “slow thinking”—is conceptually interesting but lacks a mechanistic explanation or connection to established findings in neural or cognitive science. Providing a clearer rationale, ideally supported by formal analysis of how the proposed weighted-embedding feedback influences model expressivity, optimization dynamics, or the compute–performance trade-off, would considerably strengthen both the motivation and the overall argument.

2. To substantiate the claim of a latent thinking process, it would be valuable to analyze the model’s internal states and compare them with those from explicit (e.g., CoT) and implicit thinking methods.

3. Several evaluated tasks appear susceptible to data contamination [1]. It would strengthen the paper to quantify the extent of contamination and disentangle its contribution to the observed gains—for example, by applying contamination checks, re-running on decontaminated splits, or reporting performance deltas with/without potentially contaminated items.


References

1. Koala: An Index for Quantifying Overlaps with Pre-training Corpora. Vu et al. 2023

**Questions:**

None

---

> ### Author Response · Authors · 2025-11-25
> **Response to Reviewer w4mh (Part 1/3)**
>
> Thank you for highlighting the elegance of our differentiable, self-supervised mechanism and its ability to emerge without RL or curated supervision. Below, we provide point-by-point responses to your comments.
>
>
> >**[W1]** The paper’s motivation and theoretical foundation.
>
> We formally frame PonderLM as a **Mean-Field Approximation of a Lookahead-Augmented State**. While we have developed a rigorous theoretical analysis demonstrating PonderLM's effectiveness, we prioritized extensive experimental validation in this submission due to the 9-page constraint. Nevertheless, we present the detailed theoretical framework below, which establishes: (1) why a single pondering step is mathematically superior to a standard LM, and (2) why the benefits of multi-step pondering naturally diminish, consistent with our empirical findings.
> ### 1. Why Pondering is Effective: Mean-Field Approximation
>
> Standard LMs are computationally "myopic": they compute the **current last-layer hidden state** $h_n$ based on the input sequence up to step $n$. Formally, $h_n = \mathrm{TF}(e(x_1), ..., e(x_n)) = \mathrm{TF}(c, e(x_n))$, where $e(x_i)$ retrieves the token embedding of the token $x_i$, $c$ represents the preceding context $e(x_1),..., e(x_{n-1})$, and $\mathrm{TF}(\cdot)$ denotes the Transformer backbone.
>
> However, recent advancements in reasoning (e.g., Tree-of-Thoughts, Monte Carlo Tree Search) suggest that conditioning the current state on potential future trajectories significantly improves performance. Similarly, an ideal language model with infinite compute could perform a 1-step lookahead: sampling potential next tokens from the predicted distribution $p_n$, processing them through the Transformer to observe the resultant state, and aggregating this "future information" back into the current decision process.
>
> Let $X_{n+1}$ be a random variable representing the **next token** sampled from the distribution $p_n$. We define the **Lookahead-Augmented State** generally as:
>
> $$
> \bar{h}\_n^+ := \mathbb{E}\_{X\_{n+1} \sim p\_n} [\mathrm{TF}(c, e(x\_n), e(X\_{n+1}))] = \sum\_{v \in V} p\_n(v) \cdot \mathrm{TF}(c, e(x\_n), e(v))
> $$
>
> where $p_n(v)$ is the probability of token $v$ in the vocabulary $V$.
>
> Computing this expectation exactly is intractable as it requires $|V|$ forward passes. PonderLM approximates this process efficiently via a Mean-Field Approximation. Instead of averaging the outputs (which is computationally expensive), PonderLM feeds back the expectation of the input embedding (the pondering embedding $t_n = \mathbb{E}[e(X_{n+1})]$).
>
> In our specific implementation, we integrate this "future information" via a residual connection. Consequently, we specialize the general transition function to the PonderLM architecture form: $F(z) = \mathrm{TF}(c, e(x_n) + z)$.
>
> To quantify how well PonderLM $F(\mathbb{E}[e(X_{n+1})])$ approximates the ideal state $\mathbb{E}[F(e(X_{n+1}))]$ (formulated via the same residual architecture for theoretical consistency) compared to a Standard LM, we utilize Taylor expansions.
>
> **Rationale for Expansion Points:**
> The choice of the expansion point corresponds to the operating point of each model's input.
> - **PonderLM:** The model explicitly computes and adds the mean embedding $t_n = \mathbb{E}[e(X_{n+1})]$ to the input. Thus, we expand the ideal function $\mathbb{E}[F(e(X_{n+1}))]$ around $t_n$ to quantify the discrepancy between averaging the transformer outputs (the ideal process) and feeding back the average input (the PonderLM approximation).
> - **Standard LM:** The model does not add any "future information", effectively assuming a zero-vector addition ($z=0$). Thus, we expand around $0$ to quantify the discrepancy between averaging the transformer outputs and using a zero input vector (the Standard LM baseline).
>
> We analyze the approximation error for both cases:
>
> **Case A: PonderLM (Expansion around $t_n$)**
> Expanding the function $F$ around its mean $t_n$:
> $$
> \begin{aligned}
> \bar{h}\_n^+ = \mathbb{E}[F(e(X\_{n+1}))] &= \mathbb{E}\left[ F(t\_n) + J\_F(t\_n)(e(X\_{n+1}) - t\_n) + \mathcal{O}(\Vert e(X\_{n+1})-t\_n\Vert^2) \right] \\\\
> &= F(t\_n) + J\_F(t\_n)\underbrace{\mathbb{E}[e(X\_{n+1}) - t\_n]}\_{0} + \mathcal{O}(\mathbb{E}[\Vert e(X\_{n+1}) - t\_n\Vert^2]) \\\\
> &= \boldsymbol{h}\_{\mathrm{ponder}} + \mathcal{O}(\mathbb{E}[\Vert e(X\_{n+1}) - t\_n\Vert^2])
> \end{aligned}
> $$
> where $h_{ponder} = F(t_n)$ is the actual state computed by PonderLM.
>
> **Case B: Standard LM (Expansion around 0)**
> Expanding the function $F$ around 0:
> $$
> \begin{aligned}
> \bar{h}\_n^+ = \mathbb{E}[F(e(X\_{n+1}))] &= \mathbb{E}\left[ F(0) + J\_F(0)(e(X\_{n+1}) - 0) + \mathcal{O}(\Vert e(X\_{n+1})\Vert^2) \right] \\\\
> &= F(0) + J\_F(0)\mathbb{E}[e(X\_{n+1})] + \mathcal{O}(\mathbb{E}[\Vert e(X\_{n+1})\Vert^2]) \\\\
> &= \boldsymbol{h}\_{\mathrm{std}} + \boldsymbol{J\_F(0)t}\_n + \mathcal{O}(\mathbb{E}[\Vert e(X\_{n+1})\Vert^2])
> \end{aligned}
> $$
> where $h_{std} = F(0)$ is the state computed by a Standard LM.

---

> ### Author Response · Authors · 2025-11-25
> **Response to Reviewer w4mh (Part 2/3)**
>
> **Error Analysis and Comparison:**
> We now formally compare the approximation error (distance from the ideal lookahead state $\bar{h}_n^+$) for both models. For PonderLM, the error is dominated by second-order variance terms:
>
> $$
> \Vert \bar{h}\_n^+ - h\_{ponder} \Vert \approx \mathcal{O}(\mathbb{E}[\Vert e(X\_{n+1}) - t\_n\Vert^2])
> $$
> For the Standard LM, the error contains a significant first-order linear term:
> $$
> \Vert \bar{h}\_n^+ - h\_{std} \Vert \approx \Vert \boldsymbol{J\_F(0)t}\_n + \mathcal{O}(\mathbb{E}[\Vert e(X\_{n+1})\Vert^2]) \Vert
> $$
> **Conclusion**: Since $\Vert \boldsymbol{J\_F(0)t\_n} \Vert > 0$, the Standard LM suffers from a systematic **First-Order Linear Bias**. PonderLM eliminates this bias completely by shifting the operating point to $t\_n$. This proves that $h\_{ponder}$ is mathematically closer to the ideal lookahead state $\bar{h}\_n^+$ than $h\_{std}$.
>
> ### 2. Derivation of Step-wise Corrections
>
> To understand the dynamics of multi-step pondering, we define the hidden state approximation at step $s$ as:
>
> $$
> h\_n^{(s)} := F\big(t\_n^{(s)}\big) = \mathrm{TF}\big(c,e(x\_n) + t\_n^{(s)}\big)
> $$
>
> where $t\_n^{(s)}$ is the pondering embedding generated at step $s$. We analyze the marginal gain introduced at each step via Taylor expansions.
>
> **First Pondering Step:**
> We expand $F\big(t\_n^{(1)}\big)$ around the initial state $t\_n^{(0)}$ (where $t\_n^{(0)} = \mathbf{0}$, representing the baseline LM):
>
> $$
> F\big(t\_n^{(1)}\big) \approx F\big(t\_n^{(0)}\big) + J\_F\big(t\_n^{(0)}\big)\\big(t\_n^{(1)} - t\_n^{(0)}\big) + \mathcal{O}\big(\Vert t\_n^{(1)} - t\_n^{(0)}\Vert^2\big)
> $$
>
> Rearranging this, the change in the hidden state (the first-order approximation) is:
>
> $$
> h\_n^{(1)} - h\_n^{(0)} \approx J\_F\big(t\_n^{(0)}\big)\\big(t\_n^{(1)} - t\_n^{(0)}\big)
> $$
>
> The corresponding first-order correction magnitude is:
>
> $$
> \Vert\Delta h\_n^{(1)}\Vert_{\text{(1st)}} \approx \big\Vert J\_F\big(t\_n^{(0)}\big)\\big(t\_n^{(1)} - t\_n^{(0)}\big)\big\Vert= \big\Vert J\_F\big(0\big)\\big(t\_n^{(1)} \big)\big\Vert
> $$
>
> **Second Pondering Step:**
> Analogously, we expand $F\big(t\_n^{(2)}\big)$ around the updated embedding $t\_n^{(1)}$ to find the hidden state at step 2:
>
> $$
> F\big(t\_n^{(2)}\big) \approx F\big(t\_n^{(1)}\big) + J\_F\big(t\_n^{(1)}\big)\\big(t\_n^{(2)} - t\_n^{(1)}\big) + \mathcal{O}\big(\Vert t\_n^{(2)} - t\_n^{(1)}\Vert^2\big)
> $$
>
> The hidden state variation is approximated as:
>
> $$
> h\_n^{(2)} - h\_n^{(1)} \approx J\_F\big(t\_n^{(1)}\big)\\big(t\_n^{(2)} - t\_n^{(1)}\big)
> $$
>
> Thus, the first-order term counteracted by the second pondering step is:
>
> $$
> \Vert\Delta h\_n^{(2)}\Vert_{\text{(1st)}} \approx \big\Vert J\_F\big(t\_n^{(1)}\big)\\big(t\_n^{(2)} - t\_n^{(1)}\big)\big\Vert
> $$
>
> **Generalization:**
> By generalizing this derivation to any step $s$, the marginal gain in the hidden state representation is directly proportional to the magnitude of the change in the pondering embedding:
>
> $$
> \text{Gain}\_{s+1} \approx \big\Vert J\_F(t\_n^{(s)}) \cdot (t\_n^{(s+1)} - t\_n^{(s)}) \big\Vert
> $$
> ### 3. Consistency with Empirical Results
>
> To validate this theoretical insight, we monitored the magnitude of the embedding updates, $\Vert \Delta t \Vert = \Vert t\_n^{(s)} - t\_n^{(s-1)} \Vert$, across 10 pondering steps in our pretrained Pythia-70M model.
>
> The data shows a rapid decay in the update magnitude:
>
> | Pondering Step ($s$) | 1 | 2 | 3 | 4 | 5 | 6 | 7 | 8 | 9 | 10 |
> | :--- | :---: | :---: | :---: | :---: | :---: | :---: | :---: | :---: | :---: | :---: |
> | **Magnitude $\Vert \Delta t \Vert$** | 2.453 | 1.331 | 0.903 | 0.692 | 0.580 | 0.500 | 0.432 | 0.387 | 0.360 | 0.336 |
>
> As the pondering process iterates, the delta between consecutive embeddings shrinks significantly ($2.45 \to 1.33 \to \dots \to 0.34$). According to the marginal gain formula derived previously, as $\Vert t\_n^{(s+1)} - t\_n^{(s)} \Vert$ gradually decreases, the correction applied to the hidden state also diminishes. This trend is entirely consistent with the diminishing returns observed in our experiments (Section 4.2). As shown in Figure 7 (Top), the first few pondering steps yield significant performance improvements (sharp drop in loss), while subsequent improvements gradually taper off.

---

> ### Author Response · Authors · 2025-11-25
> **Response to Reviewer w4mh (Part 3/3)**
>
> > **[W2]** To substantiate the claim of a latent thinking process, it would be valuable to analyze the model’s internal states and compare them with those from explicit (e.g., CoT) and implicit thinking methods.
>
> To alleviate your concerns, we first clarify the conceptual distinctions between our method, CoT, and implicit thinking methods, followed by case studies visualizing our model's internal states.
> 1. Distinction from CoT:
> Our method is orthogonal to Chain-of-Thought (CoT). While CoT generates extensive reasoning chains at the question level—often far exceeding the length of our pondering steps—PonderLM invokes a concise pondering process before predicting each individual token. Furthermore, unlike CoT, which collapses the probability distribution into a discrete token at every step, our method utilizes a weighted sum of embeddings. This allows the model to maintain a continuous representation of all future possibilities throughout the pondering process.
> 2. Distinction from Implicit Thinking Methods:
> Regarding implicit thinking methods, approaches like Pause Tokens merely append a fixed token, while hidden-state-based methods (e.g., Looped Transformers) lack direct interpretability. In contrast, PonderLM’s intermediate probability distributions and top candidate tokens offer a potential interpretable view of the model's inference dynamics.
> 3. Case Studies Analysis:
> Figure 3 and the additional examples in **Appendix I**, all drawn from PonderPythia-2.8B, illustrate the pondering process. We observe that the model initially considers topic-adjacent concepts—contextually relevant but often imprecise—before converging on the correct result. Following is a representative example:
>
> Inputs: *The chemical symbol for silver is*
>
> | Top 3 tokens\output probs  | Pondering steps 1 | Pondering steps 2 | Pondering steps 3 | Final predicted probs  |
> |----------------------------|-------------------|-------------------|-------------------|------------------------|
> | Rank 1                     | symbol (0.47)     | atoms (0.39)      | **Ag** (0.40)         | **Ag** (0.94)              |
> | Rank  2                    | symbols (0.32)    | atomic (0.32)     | symbol (0.33)     | ” (0.04)               |
> | Rank  3                    | nickname (0.20)   | elemental (0.30)  | elements (0.27)   |  S (0.02)              |
>
>
> - Early Steps: The model focuses on structural and broad semantic associations (e.g., predicting "symbol" or "nickname").
> - Intermediate Steps: It narrows the domain to specific chemical properties (e.g., "atoms", "elemental").
> - Final Steps: The model successfully retrieves the specific factual knowledge, with the probability of "Ag" rising significantly (e.g., from 0.40 to 0.94) to become the final output.
>
> > **[W3]** Several evaluated tasks appear susceptible to data contamination [1]. It would strengthen the paper to quantify the extent of contamination and disentangle its contribution to the observed gains—for example, by applying contamination checks, re-running on decontaminated splits, or reporting performance deltas with/without potentially contaminated items.
>
> We would like to clarify that our experiments were specifically designed to ensure a strictly fair comparison by training models on **identical data splits**:
> - Large-Scale Comparisons (Table 1, Figures 1 & 5): We utilized the **exact same pretraining dataset** (The Pile) as the official Pythia models.
> - Scaling Curves & Ablation Studies (Figure 4, Table 2): All models in these experiments were trained from scratch **using strictly identical data subsets**.
> Since the training data is controlled and identical across all comparisons, **any potential data contamination would affect both the baseline and our model equally**. Furthermore, the consistent performance improvements observed across all tasks further corroborate that the gains are driven by our method rather than data leakage.

---

### Official Review · Reviewer_Nkod · 2025-10-29

**Soundness:** 3
**Presentation:** 3
**Contribution:** 2
**Rating:** 6
**Confidence:** 4

**Summary:**

This paper takes the standard output token probabilities and feeds them back into the model a set number of time with gradients attached. The intuition is that this allows the model to "think" more before outputting a token.

**Strengths:**

Strong consistent results, good that it only needs general corpus data, comprehensive set of experiments.

**Weaknesses:**

**W1.** Limited novelty - a very simple change and very similar to prior methods. But perhaps this is not a weakness as the results seem good.

**W2.** 4x compute at inference time. With LLMs actually being used now, inference cost is important. I think they should perhaps therefore be compared to 4x larger models which will have the same inference cost. In this case the performance is less strong.

**Questions:**

**Q1.** What is meant by: “potentially reducing communication costs at scale”?

**Q2.** It is unusual for results to be quite this consistent. Are the authors sure there were no other changes that contributed to this?

---

> ### Author Response · Authors · 2025-11-25
> **Response to Reviewer Nkod**
>
> Thank you for highlighting our strong, consistent results and the efficiency of requiring only general corpus data. Below, we provide point-by-point responses to your comments.
>
> > **[W1]** Limited novelty - a very simple change and very similar to prior methods. But perhaps this is not a weakness as the results seem good.
>
> Our method is fundamentally distinct from prior approaches (e.g., Looped Transformers, Pause Tokens). Unlike methods that rely on fixed tokens or raw hidden states, we introduce a probability-weighted sum of token embeddings (pondering embedding). In Section 4.1 (Table 2), we directly compare against these baselines and demonstrate that our specific design significantly outperforms them.
>
> While the final method appears simple, it is actually the result of extensive exploration and rigorous ablation studies involving numerous complex variations. During development, we investigated various alternative designs, including:
> - Using hidden states or fixed token embeddings instead of pondering embeddings.
> - Adding MLP projectors after the pondering embedding.
> - Replacing residual connections with learnable weights.
> - Computing pondering embeddings from intermediate hidden states.
>
> Although some complex variations (e.g., learnable weights or intermediate hidden states's pondering embeddings) yielded marginal gains, we prioritized the current straightforward design to ensure generalizability and ease of adoption across different domains.
>
>
> > **[W2]** 4x compute at inference time. With LLMs actually being used now, inference cost is important. I think they should perhaps be compared to 4x larger models which will have the same inference cost.
>
> We have considered this comparison in our evaluation. In Table 1, we compare our PonderPythia-2.8B against the official Pythia-12B—a model that is more than 4× larger. Empirically, our 2.8B model surpasses Pythia-12B on multiple downstream tasks and achieves comparable average performance in both 0-shot and 5-shot settings, demonstrating that our method remains highly competitive even when compared with a model that is 4× larger.
>
> Furthermore, it is crucial to distinguish between theoretical FLOPs and actual wall-clock latency.  Modern hardware is increasingly compute-efficient, making communication bandwidth—rather than computation—the primary bottleneck for large models. A standard model that is 4× larger typically requires model parallelism (such as tensor parallelism or pipeline parallelism) to fit into memory, which introduces substantial communication overhead. PonderLM maintains a small parameter footprint ($1/4$ the size), allowing it to run on fewer GPUs (or a single device) without complex parallelism, significantly reducing this overhead. This advantage becomes even more critical at massive scales (e.g., comparing a 250B Ponder model with a 1T dense model), where communication costs grow more rapidly.
>
> Finally, maintaining a smaller parameter footprint substantially reduces GPU memory requirements for model weights (and optimizer states during training), thereby lowering the hardware barrier for deploying high-performance models.
>
> > **[Q1]** What is meant by: “potentially reducing communication costs at scale”?
>
> By "reducing communication costs at scale," we refer to the advantage gained by replacing a large, communication-heavy model with a smaller, compute-dense PonderLM.
>
> For instance, consider training in half-precision on a 48GB RTX 6000 Ada GPU. Training a Vanilla Pythia-12B would necessitate Model Parallelism (Tensor Parallelism or Pipeline Parallelism) because the model and optimizer states exceed the memory of a single GPU, incurring significant communication overhead. In contrast, a PonderPythia-2.8B model--which achieves performance comparable to the 12B baseline--can fit entirely onto a single card, thereby eliminating the need for model splitting and saving massive amounts of communication overhead.
>
> As noted in our introduction, this advantage becomes increasingly critical as model sizes scale, given that communication costs typically grow super-linearly with parameter count.
>
>
> > **[Q2]** It is unusual for results to be quite this consistent. Are the authors sure there were no other changes that contributed to this?
>
> To rigorously rule out confounding factors, we adhered to a strict experimental protocol: our PonderPythia models were trained from scratch using exactly the same training data, tokenizer, architectural components, and hyperparameters as the official Pythia suite.
>
> Throughout our experiments, we purposefully isolated the pondering mechanism as the sole variable. Consequently, we interpret this high degree of consistency across different scales and tasks not as an anomaly, but as strong evidence of the method's fundamental effectiveness and stability. To ensure full transparency and facilitate verification, we are committed to open-sourcing our code and model checkpoints, inviting the community to reproduce our results.

---

### Official Review · Reviewer_Zzsq · 2025-10-30

**Soundness:** 3
**Presentation:** 3
**Contribution:** 3
**Rating:** 6
**Confidence:** 3

**Summary:**

This paper proposes a pondering process to replace the standard single forward pass in transformers. Instead of producing a predictive distribution in one pass, the model refines it iteratively. At each iteration, the model computes the pondering embeddings by using the current predicted distribution to take a weighted sum over all token embeddings. This pondering embedding is then added back to the original token embeddings through a residual connection, and the updated embeddings are fed back into the model to produce a refined output distribution.

The pondering method shows strong empirical performance across extensive experiments. The pondering-trained Pythia models reach the same performance with significantly fewer parameters and training data, and they consistently outperform their counterparts on 9 downstream benchmarks. Ablation studies on alternative embedding strategies and the number of pondering steps further demonstrate the effectiveness of the method.

**Strengths:**

The idea is simple yet effective, and it does not rely on any external supervision. The proposed mechanism is easy to implement and can be plugged into standard Transformer architectures with minimal changes.

The empirical results are solid and sufficiently demonstrate the effectiveness of the proposed approach. The observation that performance improves monotonically with more pondering steps suggests that the method provides a controllable way to trade compute for performance.

**Weaknesses:**

Since the method introduces additional iterative passes beyond the standard forward pass, it incurs non-trivial training and inference cost, which may become particularly expensive for larger models and longer sequences.

**Questions:**

1. In Eq. (5) you add the pondering embedding via a residual connection. What happens if this residual connection is removed, i.e., if you simply set $E^1=T$. Did you run this ablation? It would help isolate the contribution of the residual pathway itself.

2. Have you examined how the predicted distribution evolves across pondering steps? It would be interesting to see whether each step moves the prediction closer to the target distribution, as this could provide additional interpretability into how the refinement process actually works.

---

> ### Author Response · Authors · 2025-11-25
> **Response to Reviewer Zzsq**
>
> Thank you for recognizing the effectiveness and plug-in nature of our method, as well as the strong empirical results. Below, we provide point-by-point responses to your comments.
>
> > **[W1]** Since the method introduces additional iterative passes beyond the standard forward pass, it incurs non-trivial training and inference cost, which may become particularly expensive for larger models and longer sequences.
>
> Although the iterative passes increase per-token computation, our method may offer superior overall efficiency by **achieving comparable performance with significantly fewer parameters**, particularly when accounting for bottlenecks like **communication overhead** and memory bandwidth.
>
> As demonstrated in Table 1, our **PonderPythia-2.8B model outperforms the official Pythia-12B**—a model more than $4\times$ its size—on multiple downstream tasks and achieves comparable average performance in both 0-shot and 5-shot settings. This suggests that PonderLM achieves higher parameter efficiency by performing more FLOPs per parameter, allowing a significantly smaller model to rival the capabilities of a massive baseline.
>
> Furthermore, this shift from parameter scaling to compute scaling aligns with the trajectory of modern hardware, which offers rapidly increasing compute power but remains bottlenecked by memory bandwidth and interconnect latency. **For larger models, the communication overhead in distributed training and inference grows super-linearly with parameter count**. By utilizing a compute-dense but parameter-light model, we can alleviate these communication bottlenecks. This potentially improves overall wall-clock efficiency despite the additional forward passes.
>
> Finally, the reduced parameter count significantly decreases GPU memory requirements for model weights and optimizer states, thereby lowering the hardware barrier for training and deploying high-performance language models.
>
> >**[Q1]** In Eq. (5) you add the pondering embedding via a residual connection. What happens if this residual connection is removed, i.e., if you simply set $E^1=T$. Did you run this ablation? It would help isolate the contribution of the residual pathway itself.
>
> We conducted an ablation study on Pythia-70M to investigate the impact of the residual connection.  We found that while the residual connection is not critical for a single pondering step, it becomes essential as the process deepens. As shown in the table below, removing the residual pathway at Steps=3 caused significant performance degradation. The residual connection seems crucial for scaling to multiple steps, possibly because it mitigates the vanishing gradient problem, consistent with standard findings in deep network training. Following are the ablation results:
>
> | Eval loss(Lower is better) | vanilla (not ponder) | Pondering Steps=1 | Pondering Steps=3|
> | -------- | -------- | -------- | -------- |
> |With Residual (Ours)|	2.83|	**2.74**|	**2.65**|
> |Without Residual|	2.83|	2.77|	2.90|
>
>
> > **[Q2]** Have you examined how the predicted distribution evolves across pondering steps? It would be interesting to see whether each step moves the prediction closer to the target distribution, as this could provide additional interpretability into how the refinement process actually works.
>
> Qualitatively, Figure 3 in the main paper presents a real-world inference example from our PonderPythia-2.8B model. We also present the process in the following:
>
> Inputs: _The longest river in the world is the_
>
>
> | Top 3 tokens\output probs | Pondering steps 1 | Pondering steps 2 | Pondering steps 3 | Final predicted probs |
> |---------------------------|-------------------|-------------------|-------------------|-----------------------|
> | Rank 1                   | river (0.60)      | Amazon (0.41)     | river (0.33)      | **Nile** (0.54)           |
> | Rank 2                   | rivers (0.36)     | river (0.35)      | Amazon (0.30)     | Amazon (0.23)         |
> | Rank 3                   | **Nile** (0.04)       | **Nile** (0.24)       | **Nile** (0.27)       | Yang (0.23)           |
>
> It explicitly visualizes how the model dynamically corrects its predictions, with the probability mass shifting closer to the target token (e.g., "Nile") at each successive pondering step. We have compiled additional examples in the Appendix I to further illustrate this step-by-step refinement.
>
> Quantitatively, we calculated the cross-entropy loss over the target labels across pondering steps using our PonderPythia-2.8B on 131072 tokens from the Pile validation set. The results are consistent with our qualitative observations: the loss decreases monotonically with each step. This confirms that each pondering step effectively moves the prediction closer to the target distribution. Following are the results:
>
> | pondering steps | s=1  | s=2  | s=3  | Final prediction |
> |-----------------|------|------|------|------------------|
> | loss            | 6.495| 6.354| 4.338| 2.365            |

---

### Author Response · Authors · 2025-12-03
**Summary of Rebuttal for the Area Chair**

Dear Area Chair,

Thank you very much for taking over the assessment of our submission. To facilitate your assessment, we first highlight the main strengths identified in the reviews, and then summarize the key concerns together with our responses.

**Key strengths highlighted by reviewers**

- **Simple, effective, and self-supervised**: Reviewers praised the method for being elegant, architecture-agnostic, and capable of emerging via standard next-token pretraining without requiring reinforcement learning or human-annotated CoT data. (Reviewers Zzsq, Nkod, w4mh)
- **Strong empirical performance**: Reviewers acknowledged the consistent and substantial gains across GPT-2, Pythia, and LLaMA architectures. Notably, PonderPythia-2.8B rivals the official Pythia-12B (4x larger), and PonderPythia-1B matches TinyLlama-1.1B (trained on 10x more data). (Reviewers Zzsq, Nkod, w4mh, We2v)
- **New scaling dimension**: Reviewers recognized the value of framing pondering as a third scaling axis (orthogonal to parameter and CoT scaling) that allows trading compute for performance in a controllable way. (Reviewers Zzsq, w4mh)

**Main questions and our responses**
- **Robustness of scaling laws**: We provided additional scaling curves for LLaMA models trained with 2 and 4 pondering steps, confirming that the scaling laws hold consistently. (c.f. response to Reviewer We2v, W1)
- **Internal dynamics and interpretability**: We added qualitative case studies (c.f. response to Reviewer w4mh, W2) and quantitative analyses (c.f. response to Reviewer We2v, Q1&Q4) of the pondering process (including Cosine similarity, Spectral analysis, KL divergence, and Cross-Entropy Loss on target tokens). These results visually and numerically confirm that the model progressively refines predictions towards the correct target. (c.f. response to Reviewer Zzsq, Q2)
- **Theoretical foundation**: We provided a formal theoretical framework modeling PonderLM as a Mean-Field Approximation of a Lookahead-Augmented State. We derived mathematically why a single pondering step reduces linear bias compared to standard LMs and why marginal gains naturally diminish with more steps, aligning with our empirical findings. (c.f. response to Reviewer w4mh, W1)
- **Comparison with larger models**:  We highlighted that PonderPythia-2.8B achieves performance comparable to Pythia-12B (4x larger), while significantly reducing  communication overhead and memory footprint in distributed settings. (c.f. response to Reviewer Nkod, W2)

We sincerely appreciate the Chair's time and effort in assessing our submission, and we are happy to provide further clarification if anything is unclear.

---

### Meta-Review · Area_Chair_W7nv · 2026-01-16

**Summary:**

This paper proposes a method for forward passing where instead of producing a distribution in a single forward pass the model refines it iteratively. Models trained with this 'pondering'-method  seem to perform better compared to similar sized models for the same amount of data and training. The idea is similar to looped transformers but still sufficiently novel and interesting.

**Reviewer Concerns:**

The evaluation is limited to very small models like GPT2 but this is understandable and still shows a promising scientific direction.
Overall the reviewers were somewhat positive with minor reservations and I share this opinion and would recommend acceptance.

**Reviewer Scores:**

n/a

---

### Decision · Program_Chairs · 2026-01-26

Accept (Poster)